# Evidence of spin and charge density waves in Chromium electronic bands

Federico Bisti [1,2,7] ✉, Paolo Settembri [1,7] ✉, Jan Minár [3], Victor A. Rogalev [2], Roland Widmer [4], Oliver Gröning [4], Ming Shi [2,5], Thorsten Schmitt [2], Gianni Profeta [1,6] & Vladimir N. Strocov [2]

The incommensurate spin density wave (SDW) of Chromium represents the classic example of itinerant antiferromagnetism induced by the nesting of the Fermi surface, which is further enriched by the co-presence of a charge density wave (CDW). Here, we explore its electronic band structure using soft-X-ray angle-resolved photoemission spectroscopy (ARPES) for a proper bulk-sensitive investigation. We find that the long-range magnetic order gives rise to a very rich ARPES signal, which can only be interpreted with a proper first-principles description of the SDW and CDW, combined with a band unfolding procedure, reaching a remarkable agreement with experiments. Additional features of the SDW order are obscured by superimposed effects related to the photoemission process, which, unexpectedly, are not predicted by the free-electron model for the final states. We demonstrate that, even for excitation photon energies up to 1 keV, a multiple scattering description of the photoemission final states is required.

The intriguing magnetic order of the Chromium metal has been the subject of several studies since decades ago[1–5]. It presents a non-magnetic phase on a body-centered cubic (BCC) lattice structure at high temperature. Below 311 K, it transforms into an antiferromagnetic (AF) simple cubic (SC) structure of CsCl-type, modulated by an incommensurate spin-density wave (SDW), along the (100) directions, involving nearly 21 unit cells[1]. It is generally accepted that this magnetic order is induced by Fermi surface (FS) nesting, which connects electron and hole pockets around the Γ and H points of the Brillouin zone (BZ), respectively. This made Chromium a prototype system to study the connection between SDW instability and the electronic structure, which is also relevant in many 3d transition metal correlated electron systems[6], possibly providing the pairing mechanism in non-conventional superconductors, like Fe-based superconductors[7–10].

However, only recently, the real space observation of the incommensurate SDW using Spin-Polarized Scanning Tunneling Microscopy has been reported, demonstrating also the coexistence of an associated charge density wave (CDW) order[11]. On the contrary, the Chromium electronic structure has been investigated in the past, probing the Cr (110) surfaces, revealing peculiar double back-folded bands induced by the SDW[12]. The Cr (100) crystal surface has been more recently reviewed[13], comparing the probed band structure with the density functional theory (DFT) calculations on the AF phase, underlining also photoemission final state effects due to the

low photon energies used (around 60 eV). From a theoretical point of view, the study of SDWs is complicated by the difficulties of first-principles DFT in describing long-range magnetic ordering. In the case of Chromium, there are significant errors in the calculation of the magnetic moments and lattice constants[14], while the SDW is not predicted to be the ground state[14–16]. As anticipated, Chromium also presents another long range order, the CDW, which is often considered to be the second order harmonic, with half of the period, of the SDW. The exact mechanism stabilizing the CDW is still under study: it was related to the FS's nesting, with nesting vector $\mathbf{q}_{CDW} = 2\mathbf{q}_{SDW}$[17], or induced by magneto-elastic coupling with SDW[1]. Moreover, an important role of strong electron correlations has also been recently theorized[11].

To unveil the competing structural, electronic, and magnetic orders in Chromium, we exploit the advantages of soft-X-ray ARPES on the Cr (100) surface, with a sharp definition of 3D electron momentum, thanks to the large probing depth and reduced final state effects. The investigation on the FS contours, from the 3D map in the BZ, brought to the clear identification of the nesting vectors of the FS, as well as the proper distinction of the bulk signal from final-state effects and surface states. DFT calculations were performed considering the different magnetic orders, including the calculation of the bands structure in the SDW phase. Through the accurate comparison between the theoretical and the experimental band structure, combined with unfolding procedures[18–24], revealed the specific fingerprints

[1]Dipartimento di Scienze Fisiche e Chimiche, Università dell'Aquila, L'Aquila, Italy. [2]Swiss Light Source, Paul Scherrer Institute, Villigen, PSI, Switzerland. [3]New Technologies Research Centre, University of West Bohemia, Pilsen, Czech Republic. [4]nanotech@surfaces Laboratory, EMPA, Swiss Federal Laboratories for Materials Science and Technology, Duebendorf, Switzerland. [5]Center for Correlated Matter and School of Physics, Zhejiang University, Hangzhou, China. [6]CNR-SPIN L'Aquila, L'Aquila, Italy. [7]These authors contributed equally: Federico Bisti, Paolo Settembri. ✉e-mail: federico.bisti@univaq.it; paolo.settembri@graduate.univaq.it

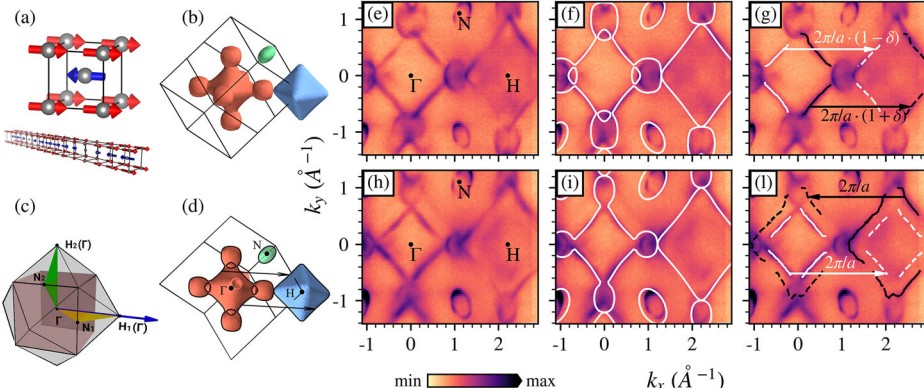

**Fig. 1 | Fermi surface nesting. a** Conventional Cr SC unit cell in an AF configuration and real space representation of the longitudinal SDW system, where the atomic magnetic moments (blue and red arrows) are modulated along the SDW direction. **b** LDA DFT calculated FS of non-magnetic BCC Cr with its BZ and (**d**) the same sliced along the plane passing through Γ, H and N points of the BZ with the nesting vectors. **c** Light grey (pale pink) the BZ of the BCC cell (SC AF cell); the blue arrow represents the SDW propagation direction; the yellow and green triangles represent the two non-equivalent paths of the SDW system. High-symmetry points of the BCC BZ are marked, with the correspondence of H and Γ for the SC cell explicitly mentioned. **e, h** Iso-energy ARPES maps at 0 eV and −0.2 eV along the plane parallel to the surface using 638 eV and circular polarization. **f, i** The same

ARPES maps are shown together with the DFT iso-energy contours at 0 and −0.2 eV of an LDA calculation of the non-magnetic BCC cell. **g** Highlight on the nesting vectors connecting the electron pocket at Γ, where the contour at the Fermi level is fitted and reported as solid black and white lines, with the hole pocket at H, where the dashed lines are replica of the solid ones translated by the related nesting vector $2\pi/a \cdot (1 \pm \delta)$ with $\delta \sim 0.05$. **l** Highlight at −0.2 eV of the AF replica (dashed white lines) of electron pocket at Γ (solid white lines) inside the hole pocket at H point (solid black lines), and the replica of the latter (dashed black lines) at Γ. The solid lines are from data fitting, and the dashed lines are their replica translated by the vector $2\pi/a$. All the experimental data are reported using a square-root color scale.

of the SDW order in the ARPES spectra, like the folded bands and gap openings close to the Fermi surface. Finally, we detected replicas in the FS ARPES signal along the plane perpendicular to the crystal surface. Rather than being connected to the magnetic order, they can be only be explained within one step-photoemission calculations, considering a multitude of final-state Bloch waves with different out-of-plane momenta. Therefore, surprisingly, the free electron approximation does not correctly describe the final states, even for photon energies around 1 keV.

## Results

### Antiferromagnetic order and nesting in the BCC band structure

The simplest model of the Cr magnetic phase is to ignore the long-range order, and approximate it with a simple AF unit cell, as depicted in Fig. 1a. In addition, we also report in Fig. 1a, the unit cell in the case of AF ordering together with a sketch of the longitudinal SDW present at low temperature. In panel (c) the Brillouin zone (BZ) in the absence or presence of the AF order is shown. In panel (b), the FS of non-magnetic BCC Cr obtained from a DFT computation is presented. It consists of two octahedrals: the first (in red) around the Γ point is an electron pocket, while the other (in blue) around H is a hole pocket. These structures have a slightly different shape and are connected by spherical electron pockets (also in red). The Fermi surface includes ellipsoids around the N points (in green), of which only one is shown for clarity. A cut of the FS along the plane passing along the high-symmetry points Γ, H and N is presented in panel (d) of Fig. 1, which also corresponds to the cut probed in the panels (e, f, g). The Fermi surface of Cr (100) collected using 638 eV photons, chosen to correspond to Γ of the BCC BZ, is reported in Fig. 1e: circular shaped electron pockets connect the square shaped electron pocket around Γ and the square shaped hole pocket around the H points of the BCC BZ; small oval shaped hole pockets are around N points. The size of the hole pocket is slightly larger than the electron one which gives origin to two different nesting vectors, $2\pi/a \cdot (1 \pm \delta)$ with $\delta \sim 0.05$. Such obtained value for $\delta$ agrees with the one measured by high-resolution X-ray scattering of bulk Cr[25]. Apart from an overestimation of the hole pockets around the N point, the other main features are well described within a simple LDA-DFT calculation in the non-magnetic phase as shown in Fig. 1f. The nesting conditions can be identified on the FS; in Fig. 1g we report in white (black) line the left (right) side of the electron pocket at Γ as fitted by momentum dispersive curve (MDC) analysis (using

Lorentzian line shape functions with second order polynomials for the background), and the same data are then rigidly translated by the nesting vectors $2\pi/a \cdot (1 \pm \delta)$, gaining a perfect overlap with the hole pocket at H. The same nesting vectors can be seen in panel (d), showcasing the 3D nesting conditions present in Cr. In the panel (h) of Fig. 1, we report the -0.2 eV iso-energy cut map for the same dataset of the Fermi surface. It is possible to distinguish new features surrounding the electron pocket at Γ, or inside the hole pocket at H, which were also barely visible on the FS. These features are related to the AF order. Of course, the LDA-DFT calculation on the non-magnetic phase is only capable of explaining the previous main features, expanded or contracted depending on the electron or hole character (see Fig. 1i). To demonstrate the AF order origin, we first have to consider that the new order reduces the BZ, making the H point of the non-magnetic phase equivalent to the Γ point of the AF one. So, the magnetic order should manifest as a band folding of the H points bands into the Γ point. The new features are indeed the relative replica of the main signals, hole, and electron pockets. In Fig. 1l the electron (hole) pocket around Γ (H) point have been fitted by MDC analysis and reported as white (black) lines, then the same data coming from the fit are translated by the reciprocal vectors $2\pi/a$ and reported as dashed lines. Also in this case, the perfect overlap of this translated data with the new features demonstrates their magnetic origin. After this initial screen on the electronic band structure, we move into a detailed examination along different directions connecting the high-symmetry points of the BZ (Γ, H, N), and compare them with DFT calculations in different magnetic phases. In Fig. 2a, g we report the experimental band structures along the Γ-H direction, obtained using $p-$ and $s-$ polarizations, while in panels (d, l) the Γ-N direction is reported. In the (b, e, h, m) panels, we report the corresponding bands obtained from a non-magnetic LDA computation, while in the (c, f, i, n) panels the LDA+U AF bands are shown. The AF case was studied using LDA+U calculations where the U value was tuned to obtain a magnetization value of $\simeq 0.6\,\mu_B$[26] (matching the experimental value in refs. 1,27,28), since in LDA simulations it is lower than that (about 0.3 $\mu_B$); in Fig. 2 U = 0.08 eV is used.

For a proper comparison, the bands have been further projected into the symmetric or antisymmetric basis of the BCC primitive cell, simulating the selection rules of ARPES for the different light polarizations (see Supplementary Note 2 in the Supplementary Information (SI)). As for the FS, the signal with the highest intensity is well reproduced by the non-magnetic

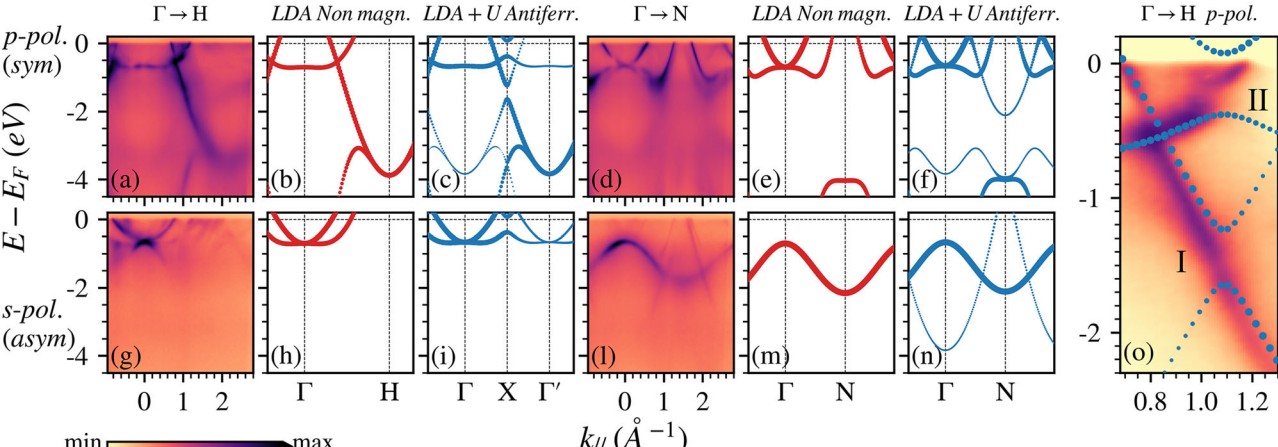

**Fig. 2 | AF features in the band structure.** (**a**), (**d**) and (**g**), (**l**) show the band structures along the different directions in reciprocal space, probed using 638 eV photons with $p(s)$-polarization, respectively. LDA band structures are shown for a non-magnetic BCC unit cell (**b**, **e**, **h**, **m**) and an AF SC cell (**c**, **f**, **i**, **n**), decomposed according to the corresponding symmetry. **o** shows a zoom of the ARPES data from (**a**), combined with the band structure of (**c**), to highlight the gaps present along the Γ-H direction. For the AF case, an LDA+U computation was performed, with U = 0.08 eV, chosen so that the atomic magnetic moment was M ≃ 0.6 $\mu_B$. The bands are unfolded onto the BCC cell to emulate the experimental ARPES intensities, and a rigid shift of +0.2 eV was adopted. The experimental data in (**a**, **d**, **g**, **l**) are reported using a square-root color scale, in (**o**), a linear color scale has been adopted instead.

phase (b, e, h, m). However, it is possible to recognize gap openings along the Γ-H direction, at the boundary of the BZ in the magnetic phase (X = $\pi/a \simeq 1.1$ Å$^{-1}$), better visible in the zoom-in plot reported in Fig. 2o and indicated as I and II. Such gaps have a magnetic origin, since reproduced only in the LDA+U AF bands, but the calculations are overestimating their values with respect to the ARPES signal, even if the magnetic moment is tuned to match the experimental one. A study on the effects of the different magnetization values both on the unfolded intensities and on the band gaps of the band structure is reported in the Supplementary Note 2 of the SI. The next step is to compare the experimental data with the calculations in which the SDW is stabilized.

### Spin and charge density wave coexistance

To study the effect on the band structure of the formation of the SDW, we performed an unfolding procedure of the DFT band structure for an SDW supercell, into the BCC cell, and compared it to the corresponding ARPES signal. SDW is stabilized on a 21 unit cell supercell, which implies q = 0.952, with the magnetization maximum amplitude tuned at 0.6 $\mu_B$ by using U = 0.13 eV, matching the experimental measurements from the literature[27,28]. The so stabilized SDW results in non-zero forces on the atoms, caused by the broken symmetry due to the modulation in the atomic magnetic moments. By relaxing the atoms following the calculated forces, a clear sinusoidal modulation with amplitude Δ ~ 0.008 Å and a period half that of the SDW, was observed. Such relaxed atomic positions are accompanied by a charge modulation, pointing to the presence of a real CDW phase, in agreement with the experimental observations[1,29]. The calculated charge density modulation has a π phase-shift with respect to the SDW: the higher charge density is in correspondence of the nodes of the SDW, while the lower charge density is located on the atoms with the highest magnetic moment. Thus, an (out-of-phase) π phase-shift with respect to the modulation of the SDW is predicted.

The presence of the SDW makes its axis of propagation non-equivalent to the other two orthogonal axis, for this reason we studied the band structure of the system on two different paths: a path orthogonal to the SDW direction, shown in Fig. 1c as a green triangle, and a path that we refer to as parallel to the SDW direction, shown as a yellow triangle in Fig. 1c.

In Fig. 3 the experimental band structures, panels (a, d, g, l), are compared to the unfolded DFT bands along the direction perpendicular to the SDW, panels (b, e, h, m), and parallel to it, panels (c, f, i, n). Overall, the main signal is still well interpreted by the calculation along both paths.

However, we can distinguish a different behavior for the gaps openings along the Γ-H, in particular in the panels (b, c). Along the path perpendicular to the wave, the gaps are reduced with respect to the AF phase, which increases the agreement with the experimental data; this can be seen clearly in panel (o), where a zoom of the panel (a) ARPES data is combined with the band structure of panel (b). The parallel path is instead presenting a doubling of the gaps not observed in the ARPES data. This already suggests that the SDW in our sample is oriented perpendicularly to the surface, as expected for the Cr (100)[12]. To support this statement, we can also distinguish a much better agreement along the Γ-H for the panel (h) rather than (i), since all the minor signals, which were not even present in the AF phase, are well reproduced only in the former panel. Indeed, the signal between −2 eV and −0.7 eV in binding energy in panel (g) has no explanation without considering the SDW. Moving into the Γ-N direction, we notice the absence of the double back-folding near the Fermi level in the path perpendicular to the wave (e), in agreement with our experimental data (d). The double back-folding is instead reproduced along the other path (f), matching the experimental measure on (110) Cr[12] (see also Supplementary Note 3 in the SI). Finally, the panel (m) is capable of explaining the highly dispersing bands crossing the Fermi level near the N point, at around 1.1–2.1Å$^{-1}$ in the panel (l), which are not present in panel (n).

Since the unfolded bands shown in Fig. 3 are actually obtained from the system in which both SDW and CDW are stabilized, in Supplementary Note 5 of the SI we also explored how the bands would appear in the presence of CDW order alone: analog contributions from the long range modulation are recognizable, but gap openings are absent, being related to the magnetic order.

Additional theoretical and experimental band structures are reported in the Supplementary Note 3 and 4 of the SI.

### Fermi surface and final state effects

Given the consistency of the results shown by the DFT and unfolding approach, we explored the iso-energy surfaces in a plane parallel ($k_x = 0$) and a plane perpendicular ($k_z = 0$) to the SDW direction. The obtained results are shown in Fig. 4, where the (a, b) panels show the plane perpendicular to the SDW with a cut at the FS and at −0.2 eV, respectively, while the (c, d) panels show the parallel plane, with the same cuts. When examining the iso-energy curves we can see the presence of relevant features by comparing the SDW unfolded data and the non-magnetic case, also shown in Fig. 4 as black lines. In panel (a), we can identify two square contours, a small one around Γ and a large one around H; those lines are also present in the non-magnetic case, as

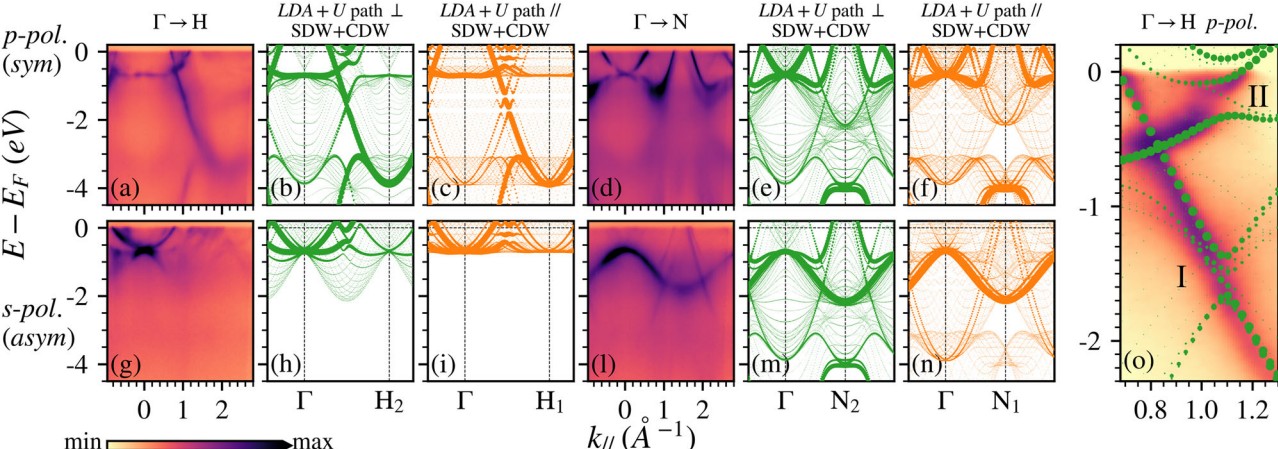

**Fig. 3 | SDW features in the band structure.** (**a, d**) and (**g, l**) panels show the band structures along the different directions in the reciprocal space probed using 638 eV photons with $p(s)$-polarization, respectively. Panels (**b, c, e, f, h, i, m, n**) show the unfolded LDA+U band structures for a 21 unit cell SDW system. In order to obtain a $M = 0.6\mu_B$ maximum magnetization of the SDW, U = 0.13 eV was adopted. (**b, e, h, m**) and (**c, f, i, n**) panels show the band structures obtained along a path that is orthogonal (parallel) to the SDW propagation direction: green (yellow) triangle in Fig. 1c. **o** shows a zoom of the ARPES data from (**a**), combined with the band structure of (**b**), to highlight the gaps present along the Γ-H direction. The bands have been decomposed according to their symmetry. A rigid shift of +0.2 eV was adopted. The experimental data in (**a, d, g, l**) are reported using a square root color scale, in (**o**) a linear color scale has been adopted instead.

it can be seen from the figure (black lines). In panel (b), where the cut at −0.2 eV is shown, two new features appear in the SDW unfolded data: these are the AF replicas of the two square contours around Γ and H, showing an AF ordering in the plane orthogonal to the SDW. These features differ in intensity, which is given by the unfolding weight: the inner lines around Γ are more intense than the outer ones, while around H, the opposite is true. This result is perfectly consistent with the ARPES data shown in Fig. 1, since, as previously mentioned, the (100) orientation of the Cr samples implies an SDW that develops in the direction perpendicular to the sample surface. In panels (c, d) of Fig. 4, in which the unfolding of the iso-energy curves in the plane parallel to the SDW is reported, we note a different behavior of the states around Γ and H. In panel (c), the square contour around Γ (H) is already accompanied by replicas of the square around H (Γ). To be more precise, there are two replicas of the states around H (Γ) exactly shifted for obtaining an overlap of a side with the main square contour around Γ (H). The proof of such perfect overlap is gained in panel (d), where, by reporting the cut at −0.2 eV, the main electron (hole) square at Γ (H) and the related two replicas at H (Γ) contract (expand), showing a clear split of the AF replicas due to the SDW order. Those split lines in the iso-energy curves, not present in the plane perpendicular to the SDW, correspond to the double back-folding visible in Fig. 3f.

Considering the adherence of the different ab-initio results and the experimental ARPES spectra, we report the experimental FS along the direction perpendicular to the crystal surface, corresponding to the direction of the SDW propagation, to compare it with the ab-initio results shown in Fig. 4c, d. The experimental results are shown in Fig. 5a; apart from the main squares around Γ (located at $k_z \sim 13.3\text{Å}^{-1}$) and H (located at $k_z \sim 15.2\text{Å}^{-1}$), the only clearly visible additional feature is a replica around H. This one is not related to the expected additional split replicas as illustrated in 4c, since it expands at lower binding energy (see Supplementary Note 4 in the SI). We attribute this replica to another effect, not related to the magnetic order but to the photoemission process itself, which is the multiple scattering effect on the photoemitted electron final state. A common approximation used in photoemission spectroscopy considers the electron as photoemitted into a free electron final state. The higher the kinetic energy of the photoemitted electron, the more this approximation is considered reasonable, since the surface potential could interfere less with such final state. However, the limit in kinetic energy beyond which this approximation is satisfied is not generically known and may certainly depend on the system under analysis. Recently, it has been shown that the free electron approximation cannot be valid even in the soft X-ray energy range[30] since replicas associated with

multiple scattering processes were distinguished in the experimental data. At the same time, it is relevant to note that such effects can be properly predicted using the one-step model of photoemission simulated spectra[30,31], as implemented within the spin-polarized fully relativistic Korringa-Kohn-Rostoker (SPRKKR) Green function framework combined with LDA-DFT[32]. In the (b, c) panels of Fig. 5, we show exactly this type of simulation for the present system. In panel (b), we report the case where a multiple scattering process is considered, and in panel (c), we show the results in the ideal case of only free electron final states. First of all, the most relevant modifications between the two treatments are the appearance of replicas between N points (along all the explored $k_z$ for a value of $k_x \sim 1.1\text{Å}^{-1}$). Such effect is exactly probed in the experimental data, and they indeed extend to photon energies greater than 1000 eV, underlying also in this case the possible limits on the free electron approximation. Then, even with very low intensity, it is possible to recognize in panel (b) the replica of the square around H. Its presence is highlighted in panel (e), where a small replica at $k_x \sim -0.6\text{Å}^{-1}$ (indicated by the red arrow) can be distinguished. Therefore, even if we were able to properly reproduce the experimental data also along the direction perpendicular to the crystal surface, we must note that these final state effects overtake the possibility of detecting the SDW order induced features. Despite the recognized limitation in our data due to the experimental geometry, confirmation on the double-splitted squared replicas of Fig. 4c, d can be found in a previous work using a different crystal orientation[12]. Indeed, in their data, it is possible to exactly recognize both the features in the FS and the split ones at low binding energy. We would also like to note how the simple and correct schematic representation of the SDW back-folding proposed in their work[12], can now find its proper theoretical demonstration, which had to go through both non-obvious calculations on the SDW phase and the use of unfolding techniques to correctly assign a spectral weight to the data.

## Discussion

From the reported soft-X-ray ARPES investigation of the Cr crystal, we were able to confirm the 3D nesting condition at its FS, which stabilizes the SDW. The detected spectroscopic signatures of the magnetic order were only partially explained by the simple AF ordering, and we have also found additional replica bands associated to the multiband character of the high-energy final states. The search for a satisfactory theoretical interpretation of the SDW from first-principles calculations has brought us to test different computational frameworks that, however, always result in an AF ground state instead of the observed SDW phase. This problem has been addressed

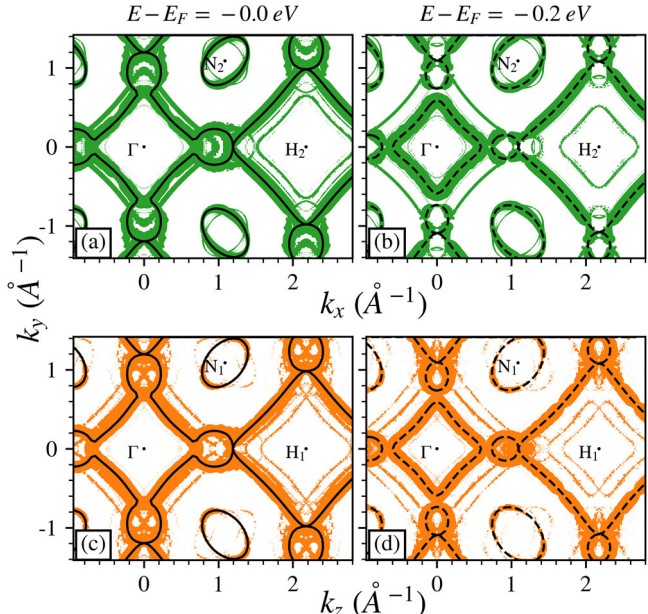

**Fig. 4 | SDW features in the Fermi surface.** Unfolded Fermi surfaces of the SDW system on a plane perpendicular to the wave direction (green dots, (**a**)) and parallel to it (orange dots, (**c**)). The FS of the non-magnetic case are also shown for reference (black lines). The iso-energy surfaces at −0.2 eV are shown in (**b, d**), using the same color scheme, with the non-magnetic case shown as black dashed lines. A 21 unit cell SDW was used to compute the Fermi surfaces. A rigid shift of +0.2 eV was adopted.

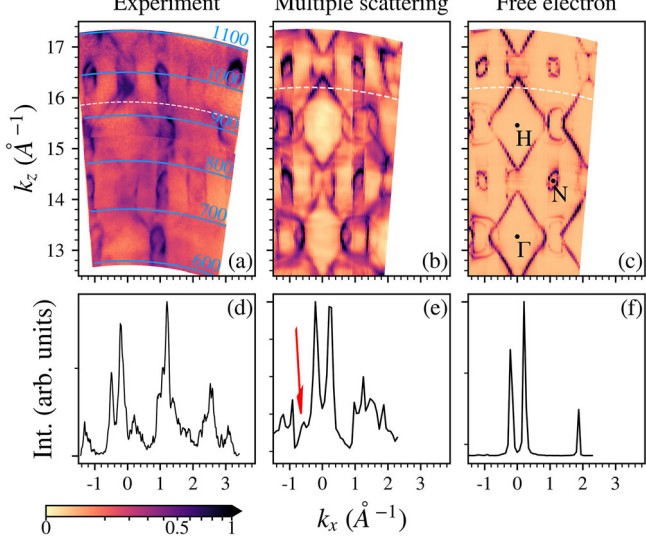

**Fig. 5 | Final state effects in the Fermi surface.** (**a**) Experimental Fermi surface along the Γ-H direction perpendicular to the crystal surface using a photon energy range of 590–1100 eV. LDA-DFT within the SPRKKR framework with multiple scattering final states, (**b**), and free-electron final states, panel (**c**). In (**d**–**f**) panels we show the intensities along the shown paths (white lines in (**a**–**c**)). All the experimental data are reported using a square root color scale.

in previous works[33], and it is a consequence of the local and semi-local approximations for the exchange-correlation functionals, which do not account explicitly for non-local AF correlations. Following refs. 14,15, we accepted to stabilize the SDW solution using LDA+U, tuning the U parameter to match the experimental magnetic moment. Interestingly, by this computational procedure, we gain a satisfactory overall agreement in the probed electronic structure, clearly accounting for the band gap openings and the prediction of back-folded bands close to the Fermi energy. Despite

being generally accepted, the band structure of the SDW has only ever been described by a basic theoretical model as a "schematic 1D representation of the backfolding of up- and down-warding parabolas"[12]. Even if this last description grasps the essence of the effect, our complete first-principles calculation accounts for all the spectrum fine details. The crucial element of our approach has been the use of the unfolding technique, capable of reproducing the photoemission interference effects by redistributing the spectral weight in the folded bands. An analog concept has been recently addressed in Ref. 34, where the treatment of a CDW as an electrostatic potential perturbation brings an eigenstates dephasing, which is then the origin of a spectral weight in the folded bands. Here, the dephasing stems from the spin unbalancing and it is described by a not straightforward unfolding from the SDW unit cell into a body-centered primitive one.

Our unfolding procedure is capable not only of highlighting the correct features in the band structure but also of explaining the intensity behavior of the folded bands. This is done by treating the magnetic order as a perturbation, the strength of which is related to the amplitude of the SDW (or the distortion amplitude of the CDW); then, it is possible to correctly predict the behavior of the unfolding weights and also map them in an analytical problem. In Supplementary Note 2 of the SI we show our results starting from the system considered in Ref. 35.

We demonstrate that once the correct SDW order is stabilized, a CDW phase naturally appears, with period and amplitude in agreement with several past experiments (see Supplementary Note 5 in the SI)[1,29,36]. Moreover, we conclude that the CDW distortion is stabilized by the SDW phase. Indeed, some theoretical papers have already proposed the stabilization of the in-phase CDW[36,37], while from the experimental side, there were controversial evidences. In fact, topographic STM images taken at low energies, about −30 meV, show a $\pi$ phase-shift of the charge density modulation period with respect to the SDW, while it goes in phase with the SDW when the images are recorded at −10 meV of bias[11]. However, it was suggested[11] that the −10 meV maps are representative of the CDW induced ordering because that energy is closer to the Fermi energy, while the $\pi$ phase-shift could originate from strong electron correlations. Our first-principles calculations consider the magneto-elastic contribution to the distortion as well as the electronic one coming from the nesting properties of the FS, with correlation effects included beyond the mean-field level, considering strong on-site Coulomb repulsion. Thus, considering that our calculations predict the $\pi$ phase-shift, it can be argued that the in-phase solution is nearly degenerate with the out-of-phase one, or it stems from residual correlations not considered by our computational approach.

Another point of discussion concerns the contextualization of our results with recent STM studies[11,38]. From our investigation, we ended up with a picture of a SDW propagating perpendicularly to the Cr (100) crystal surface, which is also in line with the known literature[5,12,39]. However, at the surface, a perpendicular commensurate SDW (C-SDW) phase (analog to the AF phase) has been observed in STM[11,38]. A good match for our data is found by considering an incommensurate SDW (IC-SDW) as shown in Fig. 3, rather than the AF configuration, Fig. 2. We did not find any signature of such surface phase. We did not detect any surface state in general, since all the detected bands are always dispersing along $k_z$. The high photon energy used in our soft-X-ray ARPES investigation well justifies the fact that we are probing the bulk properties rather than surface ones. In addition, the C-SDW phase should be characterized by a higher magnetic moment (2.6 $\mu_B$) than the bulk SDW one (0.6 $\mu_B$)[5,39], resulting in a gap opening much more pronounced than the one reported in our Fig. 3o. As we already observed, the experimental band gap can only be reproduced once the SDW order is described with the proper magnetic moment. Therefore, we can speculate that such C-SDW phase, if present also in our case, could be probably very confined at the surface, not extending much inside the crystal. The STM experiments have also probed different SDW domains on the sample surface with different propagation directions. In our measurements, the X-ray spot size (which is around 100 microns) is at least one order of magnitude greater than the size of such magnetic domains (which is around

microns). If we assume a randomly oriented bulk multi-domains scenario, our X-ray spot will average the signal over the magnetic domains. Since it is a two-over-three possibility, the signal originating from the directions perpendicular to the SDW will, therefore, weight more than the one from the parallel direction.

Finally, the physics of Cr is further enriched by important final state effects in the photoemission spectra, in particular in the direction orthogonal to the sample surface, even at photon energies up to 1 keV. This peculiar effect is also present in other materials, as recently well explained[30], making Chromium an even more complex material.

## Methods

### ARPES measurements

Cr (100) was cleaned in situ by numerous repeated cycles of Ar sputtering at 700 °C (60 min.) and annealing at 700 °C (5 min.). The photoemission measurements were performed at the soft X-ray ARPES facility[40] of ADRESS beamline[41] of the Swiss Light Source synchrotron facility, using a photon energy range of about 600-1100 eV and different light polarizations. The temperature of the samples during the measurements was around 20 K. The best combined (beamline and analyzer) energy resolution achieved in the experiments was 60–110 meV for the respective energy range of 600–1100 eV[42]. The sample surface was oriented normal to the analyzer entrance slit and the photon incidence angle was 20°.

### DFT simulations

Density functional theory (DFT) calculations were performed using the VASP code[43,44]. The Ceperly-Alder LDA exchange-correlation (xc) functional[45] was used in combination with the projector augmented wave (PAW) pseudopotential[46] for the Cr atoms, which contains 6 valence electrons, the $3d^5$ and $4s^1$ electrons (see Supplementary Note 1 in the SI for the discussion on the xc-functional choice). The energy threshold used for the self-consistent calculations was set to $10^{-8}$ eV. A kinetic energy cutoff of 400 eV for the plane wave expansion of the wavefunctions was adopted. The sampling of the Brillouin zone (BZ) has been performed using a Monkhorst-Pack uniform grid of $15 \times 15 \times 15$ $k$-points for the BCC cells and of $1 \times 15 \times 15$ for the SDW supercells; a Gaussian smearing of 0.01 eV was used. We performed DFT+U[47] calculations, varying the Hubbard term to tune the magnetic moment of the Cr atoms. Computations were performed using 2.885Å as lattice parameter. The SDWs were reproduced in DFT by creating a supercell of N different simple cubic cells stacked along an axis, as shown in Fig. 1a, and giving an initial magnetization for each atom following the profile: $m(\mathbf{r}) = m_0 \cos(\mathbf{q} \cdot \mathbf{r})$, where $\mathbf{q} = (0, 0, 2\pi\frac{q}{a})$, depends on the cell size and is chosen to be close to its experimental measures. A 21 cubic cell supercell was used, so $q$ was set to $\frac{20}{21} = 0.952$. The magnetization was treated in a collinear approach, so the calculations were not sensitive to the direction of the SDW being longitudinal or transversal.

## Data availability

The data that support the findings of this study are available from the corresponding authors upon request.

## Code availability

The codes used for the theoretical calculations and data processing are available from the corresponding authors upon a reasonable request.

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

## Acknowledgements

The research leading to these results has received funding from the Swiss National Science Foundation under Grant Agreement No. 200021_146890 and the European Community's Seventh Framework Program (FP7/2007-2013) under Grant Agreement No. 290605 (PSIFELLOW/COFUND). J.M. would like to thank the QM4ST project financed by the Ministry of Education, Youth and Sports of the Czech Republic, project no. CZ.02.01.01/00/22_008/0004572. G.P. and P.S. acknowledge support from the CINECA supercomputing center. G.P. acknowledges funding from the European Union - NextGenerationEU under the Italian Ministry of University and Research (MUR) National Innovation Ecosystem grant ECS00000041 - VITALITY - CUP E13C22001060006. F.B. acknowledges funding from the National Recovery and Resilience Plan (NRRP), Mission 4, Component 2, Investment 1.1, funded by the European Union (NextGenerationEU), for the project "TOTEM" (CUP E53D23001710006 - Call for tender No. 104 published on 2.2.2022 and Grant Assignment Decree No. 957 adopted on 30/06/2023 by the Italian Ministry of Ministry of University and Research (MUR)). M.S. acknowledges support from the National Natural Science Foundation of China (Grant No. 12350710785).

## Author contributions

F.B., M.S., and V.N.S. conceived the project and planned the experiments. F.B., R.W,. and O.G. optimized the sample cleaning procedures. F.B., V.A.R., and V.N.S. performed the soft-X-ray ARPES experiment supported by T.S., P.S., and G.P. performed the DFT calculations. F.B., P.S., and G.P. implemented the unfolding procedure. J.M. performed the first-principles ARPES calculations. F.B., P.S., G.P., and V.N.S. analysed the data. F.B. and P.S. wrote the manuscript with contributions of G.P., V.N.S. and J.M. All authors extensively discussed the results and the manuscript.

## Competing interests

The authors declare no competing interests.

## Additional information

**Peer review information** : *Communications Materials* thanks the anonymous reviewers for their contribution to the peer review of this work. Primary Handling Editors: Dawei Shen and Aldo Isidori. A peer review file is available.

