## [Transparent Peer Review file · Communications Materials]

Evidence of spin and charge density wave in Chromium electronic bands

Corresponding Author: Professor Federico Bisti

Version 0:

Decision Letter:

Dear Professor Bisti,

Thank you for submitting your manuscript, "Evidence of spin and charge density wave in Chromium electronic bands", to Communications Materials. It has now been seen by 2 referees, whose comments are appended below. You will see that while they find your work of interest, some important points are raised. We are interested in the possibility of publishing your study in Communications Materials, but would like to consider your response to these concerns in the form of a revised manuscript before we make a decision on publication.

We therefore invite you to revise and resubmit your manuscript, taking into account the points raised.

When submitting your revised manuscript, please include the following:

-A response letter with a point-by-point reply to each of the referee comments and a description of changes made. Please include the complete referee report in the response letter. Please note that the response letter must be separate to the cover letter to the editors.

-A marked-up version of the manuscript with all changes to the text in a different colored font. Please do not include tracked changes or comments. Please select the file type 'Revised Manuscript - Marked Up' when uploading the manuscript file to our online system.

-A clean version of the manuscript. Please select the file type 'Article File'.

-An updated <https://www.nature.com/documents/nr-editorial-policy-checklist.zip> Editorial Policy checklist, uploaded as a 'Related Manuscript File' type. This checklist is to ensure your paper complies with all relevant editorial policies. If needed, please revise your manuscript in response to these points. Please note that this form is a dynamic 'smart pdf' and must therefore be downloaded and completed in Adobe Reader. Clicking this link will download a zip file containing the pdf.

In the event that your manuscript is accepted we will provide detailed guidance on our journal policies and formatting. You may however wish to ensure that the manuscript complies with our house style at this stage. See our style and formatting guide (<https://www.nature.com/documents/commsj-phys-style-formatting-guide-accept.pdf>) and checklist (<https://www.nature.com/documents/commsj-phys-style-formatting-checklist-article.pdf>) for reference.

Data availability statements and data citations policy: All Communications Materials manuscripts must include a section titled "Data Availability" at the end of the Methods section or main text (if no Methods). More information on this policy, and a list of examples, is available at <http://www.nature.com/authors/policies/data/data-availability-statements-data-citations.pdf>.

- Accession codes for deposited data
- Other unique identifiers (such as DOIs and hyperlinks for any other datasets)
- At a minimum, a statement confirming that all relevant data are available from the authors
- If applicable, a statement regarding data available with restrictions
- If a dataset has a Digital Object Identifier (DOI) as its unique identifier, we strongly encourage including this in the Reference list and citing the dataset in the Data Availability Statement.

DATA SOURCES: We strongly encourage authors to deposit all new data associated with the paper in a persistent repository where they can be freely and enduringly accessed. We recommend submitting the data to discipline-specific, community-recognized repositories, where possible and a list of recommended repositories is provided at <http://www.nature.com/sdata/policies/repositories>.

If a community resource is unavailable, data can be submitted to generalist repositories such as [figshare](https://figshare.com/) or [Dryad Digital Repository](http://datadryad.org/). Please provide a unique identifier for the data (for example a DOI or a permanent URL) in the data availability statement, if possible. If the repository does not provide identifiers, we encourage authors to supply the search terms that will return the data. For data that have been obtained from publically available sources, please provide a URL and the specific data product name in the data availability statement. Data with a DOI should be further cited in the methods reference section.

Please use the following link to submit your documents:

Link Redacted

We hope to receive your revised paper within three months; please let us know if you aren't able to submit it within this time so that we can discuss how best to proceed. If we don't hear from you, and the revision process takes significantly longer, we will close your file. In this event, we will still be happy to reconsider your paper at a later date, as long as nothing similar has been accepted for publication at Communications Materials or published elsewhere in the meantime.

Please do not hesitate to contact me if you have any questions or would like to discuss these revisions further. We look forward to seeing the revised manuscript and thank you for the opportunity to review your work.

Best regards,

Dawei Shen
Editorial Board Member
Communications Materials
orcid.org/0000-0003-2402-7956

Reviewers' comments:

Reviewer #1 (Remarks to the Author):

The manuscript by F. Bisti et al. investigates the band structure of chromium (Cr) using bulk-sensitive soft-X-ray ARPES. The authors also conducted DFT calculations, considering various magnetic orders, including the incommensurate spin density wave (SDW). A detailed comparison between the theoretical and experimental band structures was made. Cr is well-known for its incommensurate SDW state driven by Fermi surface nesting. Despite extensive studies on Cr, bulk-sensitive band structure measurements have been limited. In this work, the authors successfully identified key features of the SDW in the electronic band structure, such as band folding and gap openings near the Fermi surface. Furthermore, they detected replicas in the Fermi surface ARPES signal along the plane perpendicular to the crystal surface, which can only be explained using one-step photoemission calculations. I find this study to be carefully executed, and its findings provide valuable insights into the SDW/CDW phenomena in Cr. Therefore, I recommend it for publication in Communications Materials. Below are some comments that the authors may consider:

1. The authors claims the SDW wavevector in their Cr(001) is perpendicular to the surface. In this case the SDW will be commensurate to lattice as seen in previous STM studies (similar to interlayer AFM state). The calculation in Fig. 2 seems to address this case. However, the SDW gap opening in Fig. 2(o) is not well resolved. The authors may add some discussion on it.
2. In the recent SP-STM work of Cr(001) (ref. 11), in-plane incommensurate SDW domains have been directly observed. If such in-plane domains are measured here, what features would the authors expect to see in the ARPES spectra? (or why in-plane SDW domains can be ruled out here?). The authors may add some discussion on it.

Reviewer #2 (Remarks to the Author):

Bisti et al. present a comprehensive study on the coexistence of SDW and CDW in Chromium, utilizing soft X-ray ARPES and LDA+U calculations. The authors investigate how these intertwined orders influence the material's electronic band structure, with a particular focus on the Fermi surface and the gaps that emerge along specific directions. The study introduces an unfolding technique to interpret SDW+CDW-induced features in the ARPES spectra, which, when combined with theoretical predictions, show excellent agreement. Overall, the paper provides valuable insights into band structure simulation and prediction in a prototypical system, advancing our understanding of correlated electron systems. I think the manuscript is suitable for Communication Materials, though I still have several questions:

1. The authors show a perfect match between the band structure simulation in the plane perpendicular to the SDW and the experimental data, specifically for the Γ -centered plane. Given that in this case, SDW propagates along the crystal's out-of-plane direction, could the authors confirm whether similar excellent correspondence between the ARPES spectrum and LDA+U simulation is observed on other high-symmetry planes?
2. The authors claim that the SDW is stabilized in a 21 unit cell supercell. Could the authors confirm whether the calculations shown in Fig.3 were performed using this 21 unit cell superlattice? Additionally, in Fig.4, the authors use a 14 unit cell model for simulations. Why was a different superlattice model chosen for these calculations?
3. The authors present a k_z map, which, theoretically, should be similar to the LDA calculations along the plane parallel to SDW direction (Fig. 4c). However, the authors suggest that the discrepancy at lower binding energy is due to multiple scattering effects during the photoemission process. The duplicated bands in Fig.5 appear at higher photon energies (seemingly between 800–1000 eV), while other major band data presented in the manuscript were obtained at lower photon energy (638 eV). Since final-state effects are more pronounced at lower photon energies, could the authors clarify whether these effects have been considered in the other data presented?

Additional suggestions:

1. The manuscript uses multiple terms to describe the phase shift between SDW and CDW, including " π shift", " π -phase shift", and " π phase shift". It would enhance the clarity of the manuscript if the authors could standardize this terminology throughout the text.
2. I recommend that the authors standardize the font formatting across the figures, as many of the fonts appear inconsistent, and the contrast between labels and backgrounds is not always clear. This may cause confusion for the readers and should be addressed for better readability.

Communications Materials is committed to improving transparency in authorship. As part of our efforts in this direction, we are now requesting that all authors identified as 'corresponding author' create and link their Open Researcher and Contributor Identifier (ORCID) with their account on the Manuscript Tracking System prior to acceptance. ORCID helps the scientific community achieve unambiguous attribution of all scholarly contributions. You can create and link your ORCID from the home page of the Manuscript Tracking System by clicking on 'Modify my Springer Nature account' and following the instructions in the link below. Please also inform all co-authors that they can add their ORCIDs to their accounts and that they must do so prior to acceptance.

Version 1:

Decision Letter:

Dear Professor Bisti,

Your manuscript titled "Evidence of spin and charge density wave in Chromium electronic bands" has now been seen again by our referees, whose comments appear below. In light of their advice I am delighted to say that we are happy, in principle, to publish a suitably revised version in Communications Materials.

We therefore invite you to edit your manuscript to comply with our journal policies and formatting style in order to maximise the accessibility and therefore the impact of your work.

EDITORIAL REQUESTS

* Your manuscript should comply with our policies and format requirements, detailed in our style and formatting guide (<https://www.nature.com/documents/commsj-phys-style-formatting-guide-accept.pdf>).

* Please edit your manuscript according to the editorial requests in the attached table, and outline revisions made in the right hand column. If you have any questions or concerns about any of our requests, please do not hesitate to contact me. It is important that each request be addressed in order to avoid delays in accepting your manuscript. Please upload the completed table with your manuscript files as a Related Manuscript file.

* The editorial requests table also includes a full list of the files that must be provided upon resubmission. Please upload your files according to this table.

* An updated editorial policy checklist that verifies compliance with all required editorial policies must be completed and uploaded with the revised manuscript. All points on the policy checklist must be addressed; if needed, please revise your manuscript in response to these points. Please note that this form is a dynamic 'smart pdf' and must therefore be downloaded and completed in Adobe Reader. Clicking this link will download a zip file containing the pdf.

OPEN ACCESS

Communications Materials is a fully open access journal. Articles are made freely accessible on publication. For further information about article processing charges, open access funding, and advice and support from Nature Research, please visit <https://www.nature.com/commsmat/open-access>

Please use the following link to submit your revised files:

Link Redacted

We hope to hear from you within two weeks; please let us know if the process may take longer.

Best regards,

Dawei Shen
Editorial Board Member
Communications Materials
orcid.org/0000-0003-2402-7956

REVIEWERS' COMMENTS:

Reviewer #1 (Remarks to the Author):

I have read the reply by the authors and the revised manuscript. The authors have addressed all my comments reasonably and I would recommend the publication.

Reviewer #2 (Remarks to the Author):

In their revised manuscript, the authors have addressed all my comments. I have no further suggestions to add and I recommend the manuscript for publication in Communication Materials.

Responses to Referees' comments for the paper
"Evidence of spin and charge density wave in Chromium
electronic bands"

F. Bisti,^{1,2} P. Settembri,¹ J. Minár,³ V. A. Rogalev,² R. Widmer,⁴ O.
Gröning,⁴ M. Shi,^{2,5} T. Schmitt,² G. Profeta,^{1,6} and V. N. Strocov²

¹*Dipartimento di Scienze Fisiche e Chimiche,*

Università dell'Aquila, Via Vetoio 10, 67100, L'Aquila, Italy

²*Swiss Light Source, Paul Scherrer Institute, CH-5232 Villigen PSI, Switzerland*

³*New Technologies Research Center,*

University of West Bohemia, Pilsen, Czech Republic

⁴*nanotech@surfaces Laboratory, EMPA,*

Swiss Federal Laboratories for Materials Science and Technology,

Ueberlandstrasse 129, 8600 Dübendorf, Switzerland

⁵*Center for Correlated Matter and School of Physics,*

Zhejiang University, 310058 Hangzhou, China

⁶*CNR-SPIN L'Aquila, Via Vetoio 10, 67100 L'Aquila, Italy*

Reviewer: A

The manuscript by F. Bisti et al. investigates the band structure of chromium (Cr) using bulk-sensitive soft-X-ray ARPES. The authors also conducted DFT calculations, considering various magnetic orders, including the incommensurate spin density wave (SDW). A detailed comparison between the theoretical and experimental band structures was made. Cr is well-known for its incommensurate SDW state driven by Fermi surface nesting. Despite extensive studies on Cr, bulk-sensitive band structure measurements have been limited. In this work, the authors successfully identified key features of the SDW in the electronic band structure, such as band folding and gap openings near the Fermi surface. Furthermore, they detected replicas in the Fermi surface ARPES signal along the plane perpendicular to the crystal surface, which can only be explained using one-step photoemission calculations. I find this study to be carefully executed, and its findings provide valuable insights into the SDW/CDW phenomena in Cr. Therefore, I recommend it for publication in Communications Materials. Below are some comments that the authors may consider:

Point: 1

1. The authors claims the SDW wavevector in their Cr(001) is perpendicular to the surface. In this case the SDW will be commensurate to lattice as seen in previous STM studies (similar to interlayer AFM state). The calculation in Fig. 2 seems to address this case. However, the SDW gap opening in Fig. 2(o) is not well resolved. The authors may add some discussion on it.

Answer:

We thank the Referee for putting in relation our results with the Chromium magnetic order as probed by STM, since it gives us the opportunity to improve our discussion on its complex magnetic behavior. First of all, we would consider our investigation technique closer to neutron scattering rather than STM. We used high photon energy and a micrometer-size beam-spot, so our SX-ARPES data can be considered bulk sensitive, averaging on a micrometer region. On the contrary, STM technique gives local, nanometer size, information because of its distinctive surface sensitivity. The perpendicular direction of the SDW was assumed based on the known literature for the Cr (100) crystal [1–3]. Then, in line with

that orientation, our experimental data show good consistency with the corresponding theoretical computations. Indeed, the best agreement with the experimental data, is presented in Fig. 3(o), where the band structure has been theoretically calculated along the direction perpendicular to the propagation of an incommensurate SDW (IC-SDW). For these reasons we are convinced that our interpretation, based on bulk sensitive experimental probes, is consistent with bulk-like properties of Chromium [3, 4]. From our investigations, we do not find any signature of the commensurate SDW (C-SDW) phase as probed in STM. Indeed, we did detect only SX-ARPES signal dispersing along k_z , confirming that we are not probing any surface states. In addition, the commensurate SDW (C-SDW) phase should be characterized by an higher magnetic moment than the one of the SDW ($2.6 \mu_B$) [1, 2], resulting in a gap opening much more pronounced than the one reported in our Fig. 2(o). The observed band gap can only be reproduced once the SDW order is described with the proper magnetic moment. Therefore, we believe that the commensurate SDW (C-SDW) phase, probably originating on the surface of the material, could not extend much inside the crystal. Another relevant point is that our X-ray spot size (which is around 100 microns) is at least an order of magnitude greater than the size of magnetic domains (which is around microns). In a randomly oriented bulk multi-domains scenario, our X-ray spot will average the signal over the magnetic domains. Because it is a two over three possibility, the signal originating from the direction perpendicular to the SDW will therefore weight more than the one from the parallel direction. In conclusion, the Chromium surface magnetic order is a very interesting and debated topic; yet, we believe that it is out of the scope of the present manuscript, deserving a dedicated investigation. It could be explored, for example, with nano-ARPES using low photon energy, and dedicated ab-initio calculation considering surface effects (heavily increasing the complexity of the calculations) for the study of the stabilization of such long range magnetic order at the surface. Following the Referee's suggestion we have added these considerations in the discussion section of the main text.

Point: 2

2. In the recent SP-STM work of Cr(001) (ref. 11), in-plane incommensurate SDW domains have been directly observed. If such in-plane domains are measured here, what features would the authors expect to see in the ARPES spectra? (or why in-plane SDW domains can be ruled out here?). The authors may add some discussion on it.

Answer:

As discussed in the previous point, our ARPES signal does not reflect in-plane domains at the surface. The domain size is smaller than our beam-spot so we would be averaging their signal. Following the Referee's suggestion we have added a discussion on the possible effect of SDW domains on our experiments in the main text.

Reviewer: B

Bisti et al. present a comprehensive study on the coexistence of SDW and CDW in Chromium, utilizing soft X-ray ARPES and LDA+U calculations. The authors investigate how these intertwined orders influence the material's electronic band structure, with a particular focus on the Fermi surface and the gaps that emerge along specific directions. The study introduces an unfolding technique to interpret SDW+CDW-induced features in the ARPES spectra, which, when combined with theoretical predictions, show excellent agreement. Overall, the paper provides valuable insights into band structure simulation and prediction in a prototypical system, advancing our understanding of correlated electron systems. I think the manuscript is suitable for Communication Materials, though I still have several questions:

Point: 1

1. *The authors show a perfect match between the band structure simulation in the plane perpendicular to the SDW and the experimental data, specifically for the Γ -centered plane. Given that in this case, SDW propagates along the crystal's out-of-plane direction, could the authors confirm whether similar excellent correspondence between the ARPES spectrum and LDA+U simulation is observed on other high-symmetry planes?*

Answer:

We thank the Referee for giving us the opportunity of showcasing additional data for different directions. We added a dedicated section in the SM called "S3 Additional band structures". First of all, we analyzed the Γ -P direction, as it was well explored in Ref. [3]. It is important to note that all Γ -P directions correspond to the $\{111\}$ direction, and thus the angle that it forms with any SDW direction, which lies along the $\{100\}$ direction, is always 45° . So, regardless of the crystal or SDW orientation, the band structure along the Γ -P direction is always the same. Due to our crystal surface orientation, the (100), the Γ -P direction cannot be directly probed in our experiment, as it was instead directly accessible in Ref. [3] using the (110) crystal surface orientation. Therefore, even in the absence of data from our experiments, we reported our theoretical description for comparison with the data reported in Ref. [3]. In the SM, we report an additional figure displaying the unfolded band structure along the $\Gamma - P$ direction for the different magnetic configurations. Once again, an excellent

agreement can be recognized when compared to Fig. 9 in Ref. [3]. In particular, we can see how the commensurate SDW (C-SDW) shown in Fig. 9 (a) of Rotenberg *et al.* matches our AF calculation shown in Fig. S3 (e), while the incommensurate SDW (IC-SDW) in Fig. 9 (b) matches our SDW calculation shown in Fig. S3 (f).

Secondly, we added a comparison between the experimental data and the unfolded band structure along the Γ -N direction, with both an in-plane constant photon energy spectrum and one with a k_z component. The first one corresponds to the path perpendicular to the SDW propagation direction, while the second has a component lying on the SDW direction, possibly matching both our DFT predictions.

While the antiferromagnetic replicas can be clearly seen in both ARPES maps, the features connected with the SDW cannot be easily pointed out. Additionally, in Fig. S4 (f), we can recognize replicas of the linear bands near the N point rigidly shifted along the k_z component. In fact, one replica lies closer to N, and the other one is further away from the main signal. This behavior does not match any of the possible magnetic configurations and is instead consistent with final state effects as already pointed out in Fig. 5 of the main manuscript. This discussion on the final states is now also reported in the newly added section Sec. S4 of the Supplementary Material. Finally, in Fig. S5 (h), we added the data interpolated along the out-of-plane Γ -H direction. This data should be compared with theoretical data reported in Fig. 3 (c) and/or Fig. 3 (i) of the main manuscript. Because of the method used for extracting this information, the data cannot have the same quality of the in-plane directions. The SDW should manifest as double band gaps, smaller in energy with respect to the probed one along the in-plane direction. From the data we can recognize the analogue final state effects determining replicas of the main signal rigidly shifted in k_z . Such effects are detrimental on the k_z resolution of the bands, which looks broader than the corresponding in-plane signal. In reason of the probing geometry and the final states effect, the possible detection of SDW features along the out-of-plane direction is again compromised. In conclusion, we were able to find an excellent correspondence between our theoretical band structure and the data reported in Ref. [3] for the Γ -P directions. Instead, by exploring other possible out-of-plane high-symmetry directions through the interpolation of our experimental data, we confirm that final state effects hinder the possibility of finding SDW related features.

Point: 2

2. The authors claim that the SDW is stabilized in a 21 unit cell supercell. Could the authors confirm whether the calculations shown in Fig.3 were performed using this 21 unit cell superlattice? Additionally, in Fig.4, the authors use a 14 unit cell model for simulations. Why was a different superlattice model chosen for these calculations?

Answer:

We totally understand the Referee concern, and we are happy to provide further information about this point. The SDW has been stabilized in a 21 unit cell supercell and the calculations shown in Fig. 3 have been performed on such supercell. In Fig. 4 we initially used a 14 unit cell SDW for the Fermi surface calculations, due to its high computational cost. Nonetheless, we realize how this can induce confusion in the paper, so we recomputed the Fermi surface using the 21 unit cell SDW supercell and updated Fig. 4. In the current version of the paper all SDW related data are using the 21 unit cell supercell, removing any related ambiguity.

Point: 3

3. The authors present a k_z map, which, theoretically, should be similar to the LDA calculations along the plane parallel to SDW direction (Fig. 4c). However, the authors suggest that the discrepancy at lower binding energy is due to multiple scattering effects during the photoemission process. The duplicated bands in Fig.5 appear at higher photon energies (seemingly between 800–1000 eV), while other major band data presented in the manuscript were obtained at lower photon energy (638 eV). Since final-state effects are more pronounced at lower photon energies, could the authors clarify whether these effects have been considered in the other data presented?

Answer:

We thank the Referee for this insightful comment, we realize that the distinction between the magnetic replicas and the final state effects is not a trivial task, and we should elaborate more on this. While generally it is true that higher photon energies provide reduced final state effects, and this is the reason why this work uses soft-X-ray ARPES, their behavior is very complex and not "monotonic". However, as already discussed in the previous point, it is possible to distinguish them from their behavior: the multiple scattering effects on the final states give origin to replicas of the main signal rigidly shifted along k_z . Such effects are now already discussed from the reported band structures in Fig. S4 and S5. However, to improve the comprehension of these effects, we decided to further highlight them by reporting iso-energy maps along the Γ -H direction perpendicular to the crystal surface. In the Sec. S4 of the Supplementary Material, we showcase ARPES iso-energy maps at different binding energy along the perpendicular (Fig. S5 (a-c)) and parallel (Fig. S5 (d-f)) directions to the sample surface. The multiple scattering processes give origin to a replica of the square around H present in the out-of-plane spectrum between 800 eV and 1 keV, which indeed keeps the same distance when moving in binding energy, resulting in a square rigidly shifted in k_z for all binding energies. The magnetic order folded signal, instead, must expand when the main signal contracts (the square around Γ) or vice-versa (the square around H), as already discussed in the manuscript and reported again with the dataset at 638 eV of Fig. S5. It is important to note that in the photon energy range around 638 eV photons, such multiple scattering effects are not that visible. This demonstrates that the multiple scattering effects are not that "monotonic" as it would be reasonable to

consider, and, in our considered range, do not decrease by increasing the photon energy. In our photon energy range we intercept three Γ points, that are: Γ_{638} , Γ_{870} and Γ_{1130} . Considering the observed final states effects and the fact that the overall energy resolution of the ARPES data deteriorates by going to high photon energy, it is then obvious that the best choice is the Γ point at 638 eV.

Additional suggestions

- 1. The manuscript uses multiple terms to describe the phase shift between SDW and CDW, including " π shift", " π -phase shift", and " π phase shift". It would enhance the clarity of the manuscript if the authors could standardize this terminology throughout the text.*
- 2. I recommend that the authors standardize the font formatting across the figures, as many of the fonts appear inconsistent, and the contrast between labels and backgrounds is not always clear. This may cause confusion for the readers and should be addressed for better readability.*

Answer:

We thank the Referee for his/her comment and we understand that we can improve the readability of our manuscript. Following his/her advice, we standardized the notation for the phase shift between the SDW and CDW as " π phase-shift". We improved the consistency of the fonts used in the figures and the readability of the labels where we perceived that it could help the reader.

-
- [1] T. Hänke, S. Krause, L. Berbil-Bautista, M. Bode, R. Wiesendanger, V. Wagner, D. Lott, and A. Schreyer. Absence of spin-flip transition at the cr(001) surface: A combined spin-polarized scanning tunneling microscopy and neutron scattering study. *Phys. Rev. B*, 71:184407, May 2005.
- [2] Hartmut Zabel. Magnetism of chromium at surfaces, at interfaces and in thin films. *Journal of Physics: Condensed Matter*, 11(48):9303, dec 1999.
- [3] Eli Rotenberg, B K Freelon, H Koh, A Bostwick, K Rosnagel, Andreas Schmid, and S D Kevan. Electron. *New J. Phys.*, 7:114–114, apr 2005.
- [4] Eric Fawcett. Spin-density-wave antiferromagnetism in chromium. *Rev. Mod. Phys.*, 60:209–283, Jan 1988.

Responses to Referees' comments for the paper
"Evidence of spin and charge density wave in Chromium
electronic bands"

Federico Bisti,^{1,2} Paolo Settembri,¹ Jan Minár,³ Victor A.
Rogalev,² Roland Widmer,⁴ Oliver Gröning,⁴ Ming Shi,^{2,5}
Thorsten Schmitt,² Gianni Profeta,^{1,6} and Vladimir N. Strocov²

¹*Dipartimento di Scienze Fisiche e Chimiche,*

Università dell'Aquila, Via Vetoio 10, 67100, L'Aquila, Italy

²*Swiss Light Source, Paul Scherrer Institute, CH-5232 Villigen PSI, Switzerland*

³*New Technologies Research Centre,*

University of West Bohemia, 301 00 Pilsen, Czech Republic

⁴*nanotech@surfaces Laboratory, EMPA,*

Swiss Federal Laboratories for Materials Science and Technology,

Ueberlandstrasse 129, 8600 Dübendorf, Switzerland

⁵*Center for Correlated Matter and School of Physics,*

Zhejiang University, 310058 Hangzhou, China

⁶*CNR-SPIN L'Aquila, Via Vetoio 10, 67100 L'Aquila, Italy*

Reviewer: A

The manuscript by F. Bisti et al. investigates the band structure of chromium (Cr) using bulk-sensitive soft-X-ray ARPES. The authors also conducted DFT calculations, considering various magnetic orders, including the incommensurate spin density wave (SDW). A detailed comparison between the theoretical and experimental band structures was made. Cr is well-known for its incommensurate SDW state driven by Fermi surface nesting. Despite extensive studies on Cr, bulk-sensitive band structure measurements have been limited. In this work, the authors successfully identified key features of the SDW in the electronic band structure, such as band folding and gap openings near the Fermi surface. Furthermore, they detected replicas in the Fermi surface ARPES signal along the plane perpendicular to the crystal surface, which can only be explained using one-step photoemission calculations. I find this study to be carefully executed, and its findings provide valuable insights into the SDW/CDW phenomena in Cr. Therefore, I recommend it for publication in Communications Materials. Below are some comments that the authors may consider:

Point: 1

1. The authors claims the SDW wavevector in their Cr(001) is perpendicular to the surface. In this case the SDW will be commensurate to lattice as seen in previous STM studies (similar to interlayer AFM state). The calculation in Fig. 2 seems to address this case. However, the SDW gap opening in Fig. 2(o) is not well resolved. The authors may add some discussion on it.

Answer:

We thank the Referee for putting in relation our results with the Chromium magnetic order as probed by STM, since it gives us the opportunity to improve our discussion on its complex magnetic behavior. First of all, we would consider our investigation technique closer to neutron scattering rather than STM. We used high photon energy and a micrometer-size beam-spot, so our SX-ARPES data can be considered bulk sensitive, averaging on a micrometer region. On the contrary, STM technique gives local, nanometer size, information because of its distinctive surface sensitivity. The perpendicular direction of the SDW was assumed based on the known literature for the Cr (100) crystal [1–3]. Then, in line with

that orientation, our experimental data show good consistency with the corresponding theoretical computations. Indeed, the best agreement with the experimental data, is presented in Fig. 3(o), where the band structure has been theoretically calculated along the direction perpendicular to the propagation of an incommensurate SDW (IC-SDW). For these reasons we are convinced that our interpretation, based on bulk sensitive experimental probes, is consistent with bulk-like properties of Chromium [3, 4]. From our investigations, we do not find any signature of the commensurate SDW (C-SDW) phase as probed in STM. Indeed, we did detect only SX-ARPES signal dispersing along k_z , confirming that we are not probing any surface states. In addition, the commensurate SDW (C-SDW) phase should be characterized by an higher magnetic moment than the one of the SDW ($2.6 \mu_B$) [1, 2], resulting in a gap opening much more pronounced than the one reported in our Fig. 2(o). The observed band gap can only be reproduced once the SDW order is described with the proper magnetic moment. Therefore, we believe that the commensurate SDW (C-SDW) phase, probably originating on the surface of the material, could not extend much inside the crystal. Another relevant point is that our X-ray spot size (which is around 100 microns) is at least an order of magnitude greater than the size of magnetic domains (which is around microns). In a randomly oriented bulk multi-domains scenario, our X-ray spot will average the signal over the magnetic domains. Because it is a two over three possibility, the signal originating from the direction perpendicular to the SDW will therefore weight more than the one from the parallel direction. In conclusion, the Chromium surface magnetic order is a very interesting and debated topic; yet, we believe that it is out of the scope of the present manuscript, deserving a dedicated investigation. It could be explored, for example, with nano-ARPES using low photon energy, and dedicated ab-initio calculation considering surface effects (heavily increasing the complexity of the calculations) for the study of the stabilization of such long range magnetic order at the surface. Following the Referee's suggestion we have added these considerations in the discussion section of the main text.

Point: 2

2. In the recent SP-STM work of Cr(001) (ref. 11), in-plane incommensurate SDW domains have been directly observed. If such in-plane domains are measured here, what features would the authors expect to see in the ARPES spectra? (or why in-plane SDW domains can be ruled out here?). The authors may add some discussion on it.

Answer:

As discussed in the previous point, our ARPES signal does not reflect in-plane domains at the surface. The domain size is smaller than our beam-spot so we would be averaging their signal. Following the Referee's suggestion we have added a discussion on the possible effect of SDW domains on our experiments in the main text.

Reviewer: B

Bisti et al. present a comprehensive study on the coexistence of SDW and CDW in Chromium, utilizing soft X-ray ARPES and LDA+U calculations. The authors investigate how these intertwined orders influence the material's electronic band structure, with a particular focus on the Fermi surface and the gaps that emerge along specific directions. The study introduces an unfolding technique to interpret SDW+CDW-induced features in the ARPES spectra, which, when combined with theoretical predictions, show excellent agreement. Overall, the paper provides valuable insights into band structure simulation and prediction in a prototypical system, advancing our understanding of correlated electron systems. I think the manuscript is suitable for Communication Materials, though I still have several questions:

Point: 1

1. *The authors show a perfect match between the band structure simulation in the plane perpendicular to the SDW and the experimental data, specifically for the Γ -centered plane. Given that in this case, SDW propagates along the crystal's out-of-plane direction, could the authors confirm whether similar excellent correspondence between the ARPES spectrum and LDA+U simulation is observed on other high-symmetry planes?*

Answer:

We thank the Referee for giving us the opportunity of showcasing additional data for different directions. We added a dedicated section in the SM called "S3 Additional band structures". First of all, we analyzed the Γ -P direction, as it was well explored in Ref. [3]. It is important to note that all Γ -P directions correspond to the $\{111\}$ direction, and thus the angle that it forms with any SDW direction, which lies along the $\{100\}$ direction, is always 45° . So, regardless of the crystal or SDW orientation, the band structure along the Γ -P direction is always the same. Due to our crystal surface orientation, the (100), the Γ -P direction cannot be directly probed in our experiment, as it was instead directly accessible in Ref. [3] using the (110) crystal surface orientation. Therefore, even in the absence of data from our experiments, we reported our theoretical description for comparison with the data reported in Ref. [3]. In the SM, we report an additional figure displaying the unfolded band structure along the $\Gamma - P$ direction for the different magnetic configurations. Once again, an excellent

agreement can be recognized when compared to Fig. 9 in Ref. [3]. In particular, we can see how the commensurate SDW (C-SDW) shown in Fig. 9 (a) of Rotenberg *et al.* matches our AF calculation shown in Fig. S3 (e), while the incommensurate SDW (IC-SDW) in Fig. 9 (b) matches our SDW calculation shown in Fig. S3 (f).

Secondly, we added a comparison between the experimental data and the unfolded band structure along the Γ -N direction, with both an in-plane constant photon energy spectrum and one with a k_z component. The first one corresponds to the path perpendicular to the SDW propagation direction, while the second has a component lying on the SDW direction, possibly matching both our DFT predictions.

While the antiferromagnetic replicas can be clearly seen in both ARPES maps, the features connected with the SDW cannot be easily pointed out. Additionally, in Fig. S4 (f), we can recognize replicas of the linear bands near the N point rigidly shifted along the k_z component. In fact, one replica lies closer to N, and the other one is further away from the main signal. This behavior does not match any of the possible magnetic configurations and is instead consistent with final state effects as already pointed out in Fig. 5 of the main manuscript. This discussion on the final states is now also reported in the newly added section Sec. S4 of the Supplementary Material. Finally, in Fig. S5 (h), we added the data interpolated along the out-of-plane Γ -H direction. This data should be compared with theoretical data reported in Fig. 3 (c) and/or Fig. 3 (i) of the main manuscript. Because of the method used for extracting this information, the data cannot have the same quality of the in-plane directions. The SDW should manifest as double band gaps, smaller in energy with respect to the probed one along the in-plane direction. From the data we can recognize the analogue final state effects determining replicas of the main signal rigidly shifted in k_z . Such effects are detrimental on the k_z resolution of the bands, which looks broader than the corresponding in-plane signal. In reason of the probing geometry and the final states effect, the possible detection of SDW features along the out-of-plane direction is again compromised. In conclusion, we were able to find an excellent correspondence between our theoretical band structure and the data reported in Ref. [3] for the Γ -P directions. Instead, by exploring other possible out-of-plane high-symmetry directions through the interpolation of our experimental data, we confirm that final state effects hinder the possibility of finding SDW related features.

Point: 2

2. The authors claim that the SDW is stabilized in a 21 unit cell supercell. Could the authors confirm whether the calculations shown in Fig.3 were performed using this 21 unit cell superlattice? Additionally, in Fig.4, the authors use a 14 unit cell model for simulations. Why was a different superlattice model chosen for these calculations?

Answer:

We totally understand the Referee concern, and we are happy to provide further information about this point. The SDW has been stabilized in a 21 unit cell supercell and the calculations shown in Fig. 3 have been performed on such supercell. In Fig. 4 we initially used a 14 unit cell SDW for the Fermi surface calculations, due to its high computational cost. Nonetheless, we realize how this can induce confusion in the paper, so we recomputed the Fermi surface using the 21 unit cell SDW supercell and updated Fig. 4. In the current version of the paper all SDW related data are using the 21 unit cell supercell, removing any related ambiguity.

Point: 3

3. The authors present a k_z map, which, theoretically, should be similar to the LDA calculations along the plane parallel to SDW direction (Fig. 4c). However, the authors suggest that the discrepancy at lower binding energy is due to multiple scattering effects during the photoemission process. The duplicated bands in Fig.5 appear at higher photon energies (seemingly between 800–1000 eV), while other major band data presented in the manuscript were obtained at lower photon energy (638 eV). Since final-state effects are more pronounced at lower photon energies, could the authors clarify whether these effects have been considered in the other data presented?

Answer:

We thank the Referee for this insightful comment, we realize that the distinction between the magnetic replicas and the final state effects is not a trivial task, and we should elaborate more on this. While generally it is true that higher photon energies provide reduced final state effects, and this is the reason why this work uses soft-X-ray ARPES, their behavior is very complex and not "monotonic". However, as already discussed in the previous point, it is possible to distinguish them from their behavior: the multiple scattering effects on the final states give origin to replicas of the main signal rigidly shifted along k_z . Such effects are now already discussed from the reported band structures in Fig. S4 and S5. However, to improve the comprehension of these effects, we decided to further highlight them by reporting iso-energy maps along the Γ -H direction perpendicular to the crystal surface. In the Sec. S4 of the Supplementary Material, we showcase ARPES iso-energy maps at different binding energy along the perpendicular (Fig. S5 (a-c)) and parallel (Fig. S5 (d-f)) directions to the sample surface. The multiple scattering processes give origin to a replica of the square around H present in the out-of-plane spectrum between 800 eV and 1 keV, which indeed keeps the same distance when moving in binding energy, resulting in a square rigidly shifted in k_z for all binding energies. The magnetic order folded signal, instead, must expand when the main signal contracts (the square around Γ) or vice-versa (the square around H), as already discussed in the manuscript and reported again with the dataset at 638 eV of Fig. S5. It is important to note that in the photon energy range around 638 eV photons, such multiple scattering effects are not that visible. This demonstrates that the multiple scattering effects are not that "monotonic" as it would be reasonable to

consider, and, in our considered range, do not decrease by increasing the photon energy. In our photon energy range we intercept three Γ points, that are: Γ_{638} , Γ_{870} and Γ_{1130} . Considering the observed final states effects and the fact that the overall energy resolution of the ARPES data deteriorates by going to high photon energy, it is then obvious that the best choice is the Γ point at 638 eV.

Additional suggestions

- 1. The manuscript uses multiple terms to describe the phase shift between SDW and CDW, including " π shift", " π -phase shift", and "pi phase shift". It would enhance the clarity of the manuscript if the authors could standardize this terminology throughout the text.*
- 2. I recommend that the authors standardize the font formatting across the figures, as many of the fonts appear inconsistent, and the contrast between labels and backgrounds is not always clear. This may cause confusion for the readers and should be addressed for better readability.*

Answer:

We thank the Referee for his/her comment and we understand that we can improve the readability of our manuscript. Following his/her advice, we standardized the notation for the phase shift between the SDW and CDW as " π phase-shift". We improved the consistency of the fonts used in the figures and the readability of the labels where we perceived that it could help the reader.

-
- [1] T. Hänke, S. Krause, L. Berbil-Bautista, M. Bode, R. Wiesendanger, V. Wagner, D. Lott, and A. Schreyer. Absence of spin-flip transition at the cr(001) surface: A combined spin-polarized scanning tunneling microscopy and neutron scattering study. *Phys. Rev. B*, 71:184407, May 2005.
- [2] Hartmut Zabel. Magnetism of chromium at surfaces, at interfaces and in thin films. *Journal of Physics: Condensed Matter*, 11(48):9303, dec 1999.
- [3] Eli Rotenberg, B K Freelon, H Koh, A Bostwick, K Rossnagel, Andreas Schmid, and S D Kevan. Electron. *New J. Phys.*, 7:114–114, apr 2005.
- [4] Eric Fawcett. Spin-density-wave antiferromagnetism in chromium. *Rev. Mod. Phys.*, 60:209–283, Jan 1988.